# Intelligent Evaluation of Stone Cell Content of Korla Fragrant Pears by Vis/NIR Reflection Spectroscopy

**DOI:** 10.3390/foods11162391

**Published:** 2022-08-09

**Authors:** Tongzhao Wang, Yixiao Zhang, Yuanyuan Liu, Zhijuan Zhang, Tongbin Yan

**Affiliations:** 1Agricultural Engineering Key Laboratory, Department of Xinjiang Uygur Autonomous Region, University of Education, Alar 843300, China; 2College of Mechanical and Electrical Engineering, Tarim University, Alar 843300, China

**Keywords:** successive projective algorithm, uninformative variable elimination, support vector regression, Korla fragrant pear, stone cell content, intelligent evaluation

## Abstract

Stone cells are a distinctive characteristic of pears and their formation negatively affects the quality of the fruit. To evaluate the stone cell content (SCC) of Korla fragrant pears, we developed a Vis/NIR spectroscopy system that allowed for the adjustment of the illuminating angle. The successive projective algorithm (SPA) and the Monte Carlo uninformative variable elimination (MCUVE) based on the sampling algorithm were used to select characteristic wavelengths. The particle swarm optimization (PSO) algorithm was used to optimize the combination of penalty factor C and kernel function parameter g. Support vector regression (SVR) was used to construct the evaluation model of the SCC. The SCC of the calibration set ranged from 0.240% to 0.657% and that of the validation set ranged from 0.315% to 0.652%. The SPA and MCUVE were used to optimize 57 and 83 characteristic wavelengths, respectively. The combinations of C and g were (6.2561, 0.2643) and (2.5133, 0.1128), respectively, when different characteristic wavelengths were used as inputs of SVR, indicating that the first combination had good generalization ability. The correlation coefficients of the SPA-SVR model after pre-processing the standardized normal variate (SNV) for both sets were 0.966 and 0.951, respectively. These results show that the SNV-SPA-SVR model satisfied the requirements of intelligent evaluation of SCC in Korla fragrant pears.

## 1. Introduction

Korla fragrant pears are popular with consumers worldwide due to their beautiful skin color, sweet and crisp flesh, and rich fragrance. Consumers pay close attention to appearance quality and edible quality when they purchase Korla fragrant pears. A special characteristic of pears is that various stress factors induce the formation of stone cells, composed of large amounts of lignin and cellulose, which negatively affects the edibility and quality of the fruits [1]. There is a large number of soluble cells in some Korla fragrant pears of inferior quality, such as ‘green top fruit’ and rough skin fruit [2,3]. The taste is more delicate if there is lower stone cell content (SCC) in fresh Korla fragrant pears. Up to now, research on stone cells has mostly been concerned with their content [4] and structural characterization [5,6,7] in different germplasms of pears. Moreover, traditional methods of evaluation of the quality of edible fruit are destructive and require long processing times.

Visible and near-infrared (Vis/NIR) spectroscopy is fast, safe, and contactless technology that has been used to evaluate some qualities of fruits, including soluble solids content (SSC) [8,9,10,11,12], firmness [13,14,15,16], titratable acidity [17], dry matter [18], and polyphenols to amino ratio [19]. Intelligent evaluating research about SCC has not been conducted. In addition, samples are needed to change detecting places manually, and the illuminating angles cannot be changed automatically when the Vis/NIR spectroscopy systems are used.

While Vis/NIR spectrometry usually yields thousands of response signals per detection point, not all wavelengths are associated with corresponding chemical bonds as a quality parameter of one kind of agro-product. Furthermore, the spectra at certain wavelengths do not make significant contributions to the evaluation models. Therefore, it is essential to extract characteristic wavelengths to establish intelligent models that are robust, require less computing time, and are highly predictive. Successive projective algorithm (SPA) [20,21] and uninformative variable elimination (UVE) [22,23] are two filter-type [24] algorithms that have been widely used to extract characteristic wavelengths. The SPA and UVE were evaluated using the results of correlation coefficient (R), root mean square error (RMSE), multiple linear regression (MLR), and partial least square regression (PLSR) as bases to estimate the combinations of the wavelengths. MLR [25,26] and PLSR [27,28] have different principles of establishing evaluating models. It is unreasonable to use different evaluating models to estimate the combinations of characteristic wavelengths.

A support vector machine (SVM) is a generalized linear classifier that categorizes data into two classes based on supervised learning. It has several advantages, including stability, sparsity, and simplicity. Previous applications of this algorithm mainly included qualitative analyses [29,30,31,32]. Recently, support vector regression (SVR) has been applied in quantitative evaluations as part of the development of SVM theories [33,34,35].

Taking all these factors into account, this study was conducted with three aims: (1) to set up an intelligent spectra acquisition system for Korla fragrant pears in which the optical subsystem has an adjustable irradiation angle and samples can be rotated at a specific angle; (2) to choose characteristic wavelengths to simplify the detection of SCC in Korla fragrant pears by SPA and UVE; and (3) to establish an SVR evaluating model after optimizing penalty factor C and kernel function parameter gamma (g).

## 2. Materials and Methods

### 2.1. Korla Fragrant Pears and Pretreatment

Korla fragrant pears were collected from a plantation (Alar, Xinjiang, China) from 15 to 20 September 2021. A total of 120 fruit samples were selected with uniform spindle shape (diameter 61–85 mm) and weight (110–130 g), and without visual damage on the surface.

Samples were soaked in a mixture of water and a special fruit cleaning agent (Almawin, Germany), with chemical compositions of plant sugar surfactant, citric acid, organic lemon extract, glycerin, and lactic acid, for about 30 s, and then rinsed twice with distilled water. The cleaned pears were air-dried at room temperature (20 °C) and then stored in a preservation box at 4 °C. Prior to Vis/NIR spectra acquisition, samples were placed at room temperature for 30 min. Each Korla fragrant pear was coded by a labeled paper (24 × 12 mm) which was attached to the end of the calyx.

### 2.2. Vis/NIR Spectroscopy System and Diffuse Reflectance Spectra Acquisition

The Vis/NIR spectroscopy system is shown in Figure 1. The system is composed of a spectra acquisition unit, a light source, a sample rotating unit, and a computer. The spectroscopy system is placed in the dark room.

A shortwave spectrometer (USB2000+, Ocean Optics Inc., Dunedin, FL, USA) and an optical fiber component (QP400-2-VIS-BX, Ocean Optics Inc., Dunedin, FL, USA) make up the spectra acquisition unit. The range of wavelengths is between 468 nm and 1155 nm, and the bits of A/D conversion is 12. The spectrometer sends spectra to the computer by serial communication. There is 1 receiving optical fiber in the center of the optical fiber component, and the numerical aperture is 0.22 ± 0.02 mm.

The light source unit contains 2 Halogen tungsten lamp beads (20 W, Philips). Each lamp bead is fixed in a cup-shaped container which has two connecting through-holes along the center axis and two positioning blind holes perpendicular to the central axis. Each group has two mounting plates, as shown in the partial enlargement in Figure 1, with one connecting through-hole and five positioning through-holes. The distances between each of the five positioning holes and the connecting through-hole are the same, and the angle between the two adjacent connecting holes is 15°. The angle of the light beams can be adjusted via the cup-shaped container and the mounting plate group.

The sample rotating unit consists of two rubber rollers that rotate in the same direction. The distance between the centers of the rollers is 5 mm larger than the diameter of each roller. The maximum rotation of the Korla fragrant pear samples is 120° ± 0.5°.

The distance between the circular lamp rack and the sample stage is 80 mm, and the angle of each lamp bead in the vertical direction is 37.5°. In this set-up, a circle of incident light, with a diameter of 70 mm, irradiates the upper part of the Korla fragrant pears.

Prior to the acquisition of spectral data, a Korla fragrant pear weighing 120 g ± 1 g was placed on the sample rotating unit, after which the parameters of the spectrometer were adjusted. The optical fiber probe was placed in the upper center of the Korla fragrant pear at a vertical distance of 10 mm, where the detecting radius of the sample was about 2.2 mm. The integration time, scanning times, and smoothing were set to 20 ms, 5, and 2, respectively, when the maximum reflection intensity of a standard diffuse reflection whiteboard (DR300-WS-PTFE, Ocean Optics Inc., Dunedin, FL, USA) was about 55705 counts (about 85% of the maximum value). A white reference and black reference were obtained by turning the halogen lamp on and off, respectively, when the standard diffuse reflection whiteboard was placed 10 mm away from the fiber probe.

Each sample was placed on the sample rotating stage so that the surface near the maximum diameter along the minor axis was under the optical fiber probe. The first group of spectral data for the first point was obtained at the vertically downward position of the number-marked side, and the other two groups were obtained at a rotation of 120° and 240°, respectively, along the long axis. The mean spectral values of the three points were taken as spectral data for each sample.

### 2.3. Measurement of SCC

The SCC of Korla fragrant pears was measured using a modified gravimetric method [36]. After removing the peel and core, three pieces of pulp (11× 11× 8.3 mm) were cut around the spectral data collecting points and weighed with a precision electronic balance (FA3004, Shanghai Liangping Instrument Co., Ltd., Shanghai, China). The total weight of the three pieces was recorded.

The three pieces of pulp were sealed together with a self-sealing bag and cooled for 24 h at −18 °C. Next, the pulp was thawed and homogenized in 50 mL of distilled water using a small-sized tissue smasher (FL1902, Ningbo Kajafa Electrical Technology Co., Ltd., Ningbo, China) at 22,000 r/min for 1 min. The homogenate was poured into a 1000 mL beaker (beaker A). The inner wall of the plastic container of the smasher was rinsed 2–3 times with distilled water, and the cleaning solution was added to beaker A. Next, 600 mL of distilled water was added to the mixture in beaker A, stirred with a glass rod for 1 min, and allowed to stand for about 30 s. The upper suspension in breaker A was poured out into beaker B. This process was repeated 2–3 times until there was no suspended substance. The same operation was conducted for the mixture in beaker B.

The precipitated stone cells in beaker A and beaker B were filtered through filter paper. Next, the filter paper was dried at 60–65 °C in a drying oven until the weight remained unchanged. Then, the dried stone cells were collected and weighed. The SCC of sample i was calculated as:(1)$$X_{i}=\frac{m_{\mathrm{itotal}}{\;-\;m}_{\mathrm{ifilter}}}{m_{i}}\;\times \;100\%$$
where i is the serial number for samples; X_i_ refers to the SCC; m_itotal_ refers to the total weight of filter paper and stone cells; m_ifilter_ refers to the weight of filter paper; and m_i_ refers to the weight of the selected pulp.

### 2.4. Spectral Preprocessing and Sample Set Division

The original spectra inevitably shifted and displayed background noise due to the influences of the data acquisition environment, sample size, instrument, and other factors. The stability of spectral data and the signal-to-noise ratio could effectively be improved through the reasonable use of preprocessing methods. A multiple scattering correction (MSC) was used to eliminate baseline drift [37]. The standardized normal variate (SNV) was used for the centering and calibration of the spectral data in each wavelength [38]. Several Savitzky–Golay (S-G) filters [39], with frame sizes of 3–9 and fitting orders of 1–7, were used to improve the smoothing effect. The optimal combination of frame size and fitting order was chosen according to R and RMSE values. Each spectral preprocessing method and the combinations of S-G and MSC, or S-G and SNV, were used. The preprocessed spectra were used to construct different PLSR models of SCC. The best method or combination was selected according to corresponding R and RMSE values.

The sample-set partitioning method based on the joint x-y distance algorithm (SPXY) divided the samples into a calibration set (Cs) and a validation set (Vs) where spectral data and SCC were taken as the input data. The proportion of Cs:Vs was 3:1 in this study.

### 2.5. Algorithms of Selecting Characteristic Wavelengths

#### 2.5.1. SPA

The SPA is a forward variable selection method that uses simple operations to minimize the collinearity of variables in vector space [40]. Three phases are required to select characteristic wavelengths which have the least collinearities.

First, K chains with N_max variables are created by using QR decomposition of spectral matrix ${\mathrm{Spec}}_{\mathrm{Ncal}\;\times \;K}$. The number of N_max should be between the minimum value defined by the data processor and the smaller of Ncal and K. Here, Ncal and K represent the number of samples in Cs and wavelengths, respectively.

Second, K × N_maxsets of characteristic wavelengths were selected according to the root mean square error of Vs (RMSEV). Each regression coefficient vector B of the PLSR model was calculated according to Equation (2). The RMSEV of the corresponding PLSR model was calculated according to Equation (3). The set of characteristic wavelengths with the minimum RMSEV was selected.
Specc × B = Refc(2)
(3)$$\mathrm{RMSEV}\left(j\right)=\sqrt{\frac{1}{N_{\mathrm{val}}}\sum_{i=1}^{N_{\mathrm{val}}}{{(\mathrm{Ref}}_{v}(i)\;-\;{\hat{\mathrm{Ref}}}_{v}\left(i)\right)}^{2}}$$
where Specc refers to the set of preprocessed spectral data, which has Nrows (0 < N < N_max) and S columns (0 < S < K); Refc refers to the measured values of SCC corresponding to the selected N samples in Cs; Ref_v_(i) refers to the measured value of SCC of sample i in Vs; ${\hat{\mathrm{Ref}}}_{v}\left(i\right)$ refers to the predicted SCC value calculated by selected spectral data and B.

Third, uninformative wavelengths were further eliminated according to the F-test. A correlation index was defined for each selected wavelength at the end of phase 2. The index was the absolute value of the arithmetic product of the regression coefficient and the standard deviation. The originally selected characteristic wavelengths were rearranged in descending order according to the correlation indexes. Another set of PLSR models was established with the spectral data of the first j wavelengths and SCC. Corresponding RMSEVs were calculated. The critical value, t_RMSEV_, was calculated by the inverse function of the sum distribution function for the F distribution, as shown by Equation (4), for which the significance value α was 0.25 and the degrees of freedom were the same. The wavelengths whose RMSEVs were less than t_RMSEV_ were chosen as the final characteristic ones.
(4)$$t_{\mathrm{RMSEV}\;}=\frac{\mathrm{RMSEV}(j)}{\min(\mathrm{RMSEV}(j))}$$

#### 2.5.2. UVE Combined with Monte Carlo Sampling (MCUVE) and PLSR

The informative wavelengths were selected by UVE based on the regression coefficients of PLSR models. The Monte Carlo sampling method was used to randomly select *N* kinds of sample groups. The PLSR regression coefficient vector β(j,:) was obtained from the spectra and corresponding SCC vector of the jth group. The stability value C(k) at the kth wavelength was calculated by Equation(5). Wavelengths were sorted according to the values of vector C from the largest to the smallest. Evaluating models were established by adding new spectra of one wavelength, which had a smaller stability value. The wavelengths were selected as characteristic wavelengths with the minimum value of RMSEV.
(5)$$C\left(k\right)=\frac{\mathrm{mean}(\beta \left(k,:\right))}{\mathrm{std}\left(\beta (k,:)\right)}$$
where mean(β(k,:)) and std(β(k,:)) refer to the mean coefficient and standard deviation at the kth wavelength, respectively.

### 2.6. Modeling Algorithm

The radial basis function (RBF) is a good generalization of the kernel function of SVR. The particle swarm optimization algorithm (PSO) was used to determine the optimal combination of C and g [41] in order to obtain a model with good performance. The model was evaluated by RMSEV and R.

## 3. Results

### 3.1. Statistics of SCC Measured Values

There were 90 samples in the Cs and 30 samples in the Vs. The SCC values of both sets are shown in Table 1. The SCC ranged from 0.240% to 0.657% in the Cs and from 0.315% to 0.652% in the Vs. The combinations of mean value and standard deviation (SD) were (0.486%, 0.100%) and (0.481%, 0.083%), respectively. The range of SCC in Cs covered that in Vs, which ensured the feasibility of the evaluating model. An ANOVA test was taken to check SCC values in Cs and Vs through SPSS software (Version 23, International Business Machines Corporation, Armonk, NY, USA). The *p* value was 0.008, indicating that there was a significant difference between Cs and Vs.

### 3.2. Spectral Characteristics and Different Preprocessing Methods

Spectra in the range of 498–1020 nm were considered effective, owing to the large amount of noise at both ends of the original spectrum. The effective original spectral curves of Korla fragrant pears are shown in Figure 2a. There were two reflective valleys near 680 nm and 980 nm and two reflective peaks near 550 nm and 750 nm. The spectra near 750 nm and 980 nm were related to carbohydrate content [42], and O-H [43] in the flesh of fruits. SCC had a negative correlation with carbohydrate content; therefore, the spectra at these wavelengths were indirectly related to stone cells. Spectra near 550 nm and 680 nm were related to anthocyanins and chlorophyll in the sample epidermis, respectively [44]. Korla fragrant pears with high levels of stone cells usually have green skin; therefore, SCC also had some relationship with spectra near 550 nm and 680 nm.

Using preprocessing algorithms could improve the evaluation accuracy compared with not using them. The principal component numbers of PLSR models were all 10 after different preprocessing algorithms. The optimal combination of frame size and fitting order was (7, 5) where R of the calibration set and validation set grew the largest, according to Table 2. The evaluation results of PLSR models based on different spectral preprocessing algorithms are shown in Table 3. Evaluation models based on MSC and SNV had higher Rs and lower RMSEs, while those based on S-G_(7, 5)_ had lower Rs and higher RMSEs, in Cs and Vs. The robustness of the PLSR model based on SNV was better than that of MSC according to the different values of Rs between Cs and Vs. The addition of S-G_(7, 5)_ did not improve the ability of evaluation because the combination of two-point smoothing and S-G_(7, 5)_ eliminated some effective spectral information. The model established on the basis of the SNV preprocessing algorithm achieved the best results, with R and RMSE of 0.9189 and 0.0277% in the Cs, and 0.8935 and 0.0315% in the Vs. Spectral curves based on SNV are shown in Figure 2b.

### 3.3. Characteristic Wavelengths

Fewer than 100 wavelengths were required to simplify the SCC evaluating models. The changing processes of RMSEV with different wavelength candidate subsets, which were chosen by SPA or MCUVE, are shown in Figure 3a,b, respectively. Using the combination of (SPA, PLSR) or (MCUVE, PLSR), characteristic wavelengths were selected (57 and 83, respectively). The minimum RMSEVs of SPA and MCUVE were 0.0692% and 0.0685%, respectively.

As shown in Figure 4a, 46 characteristic wavelengths distributed densely between 846.8 and 940.6 nm were obtained for the first selection method, while 10 characteristic wavelengths were scattered between 498 and 750 nm and 1 characteristic wavelength was located at 997.1 nm. Characteristic wavelengths selected by the second method were distributed mainly in the ranges of 757.7–796.7 nm, 828.3–847.5 nm, 866.8–910.3 nm, and 952.9–1006.9 nm, as shown in Figure 4b. The two methods both selected characteristic wavelengths in the range of 828.3–910.3 nm, which were correlated with the third overtone stretch of O-H and C-H functional groups.

### 3.4. SCC Evaluation Based on PSO-SVR

When the selected 57 wavelengths or 83 wavelengths were used as inputs, the optimal values of C were 2.5133 and 6.2561, respectively, as shown in Figure 5a,b. This indicated that the error tolerability produced by the first wavelength group was stricter than that produced by the second group and might result in evaluation model overfitting. The optimal values of g were 0.1128 and 0.2643 for the first and second group, respectively, indicating that the support vector number of the first wavelength group was less than that of the second. The optimal combination of C and g was obtained in the 36th iteration for MCUVE where the fitness was 0.01394%, and for SPA in the 100th iteration where the fitness was 0.01404%.

The evaluation of SCC based on SPA-POS-SVR is shown in Figure 6a. The correlation coefficients of the Cs and Vs were 0.949 and 0.928, respectively, and the RMSEs of both sets were 0.0253% and 0.0297%, respectively. The evaluation ability in the SCC range of 0.2000–0.4000% and 0.6000–0.7000% was better than that in the SCC range of 0.4000–0.7000%, with samples of Cs and Vs evenly distributed in the interval with smaller deviations. This showed that the robustness of SPA-PSO-SVR was poor and that the adaptability of the global evaluation was low.

The evaluation results of SCC based on MCUVE-POS-SVR are shown in Figure 6b. The correlation coefficients of the Cs and Vs were 0.966 and 0.951, respectively, and the RMSEs of both sets were 0.0209% and 0.0239%, respectively. The evaluation ability in the SCC range of 0.2000–0.7000% was better. Moreover, several scattering points of the Cs and Vs had relatively large deviations in the SCC ranges of 0.3000–0.4000% and 0.6000–0.7000%. Overall, the evaluation accuracy and robustness of MCUVE-PSO-SVR were better than those of SPA-PSO-SVR. The MCUVE-PSO-SVR model of SCC can be applied in online systems or portable equipment for evaluating the qualities of Korla fragrant pear.

The parameters of our spectroscopy system were set according to Korla fragrant pear. The SCC evaluating model and parameters of spectroscopy system would not be suitable for other kinds of pear, such as ‘Yali’ pear, ‘Dangsha’ pear, and so on. The minimum SCC of all samples was 0.240% in Cs and 0.315% in Vs according to Table 1. The largest evaluating errors were 8.711% and 10.845% in Cs and Vs. The evaluating precision met the requirements of application. Therefore, the minimum limit of SCC that could be detected was 0.240% by the MCUVE-PSO-SVR model.

## 4. Discussion

It has been shown previously that spectroscopy technology can be applied in quantificational evaluations of the quality parameters of agricultural products. This was a new exploration to intelligently evaluate SCC for Korla fragrant pears by Vis/NIR reflection spectroscopy. An intelligent evaluating system was introduced. Its illuminating angle could be adjusted and Korla fragrant pears could be rotated along the long axis with a set angle.

Several processes were undertaken in order to obtain an accurate and stable evaluating model of SCC. The SNV algorithm had the best evaluation effect among the spectral pretreatment methods, according to evaluating accuracy and robustness. A total of 83 characteristic wavelengths were selected by using Vis/NIR spectra combined with MCUVE algorithm. Finally, the MCUVE-PSO-SVR model was established. The correlation coefficients of the Cs and Vs were 0.966 and 0.951, and the root mean square errors of both sets were 0.0209% and 0.0239%. The results demonstrate that the model could achieve the quantitative detection of SCC in Korla fragrant pear.

The nondestructive and rapid evaluation system combined with the MCUVE-PSO-SVR model for the SCC of Korla fragrant pears could meet the requirements of detection time and evaluation precision. It could be rendered more suitable for industry work requirements through the adjustment of key parameters and the development of supporting equipment. Its transformation and applications in industry are more meaningful, and present an attractive target for future research.

## Figures and Tables

**Figure 1 foods-11-02391-f001:**
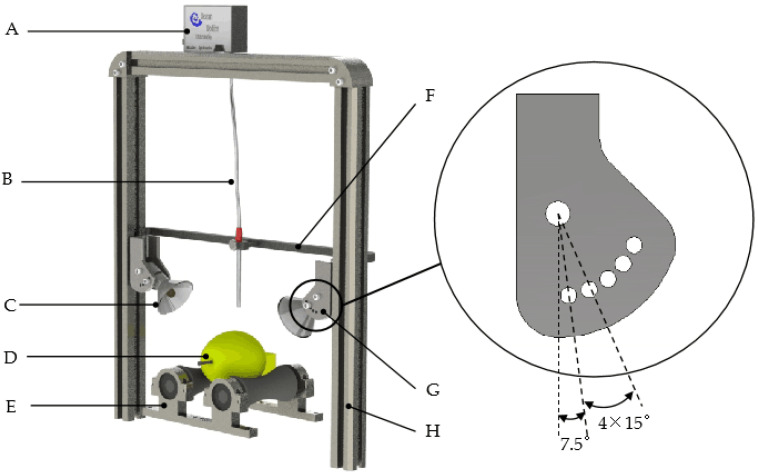
Vis/NIR spectra acquisition system for Korla fragrant pears. A: spectrometer; B: optical fiber; C: halogen lamp; D: sample; E: rotating stage; F: optical fiber bracket; G: lamp mounting plate; H: system mounting rack.

**Figure 2 foods-11-02391-f002:**
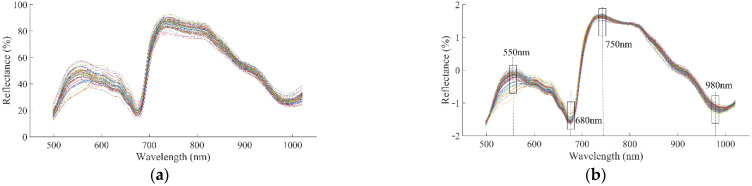
Reflective spectral curves. (**a**) Raw spectrum; (**b**) spectrum after SNV pretreating.

**Figure 3 foods-11-02391-f003:**
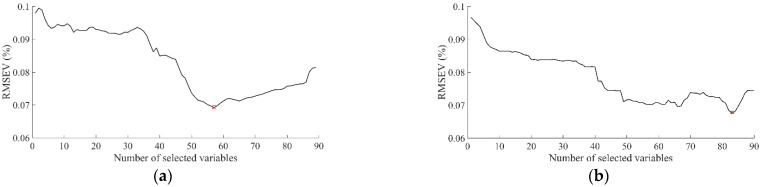
Changing processes of RMSEV with different wavelengths. (**a**) SPA; (**b**) MCUVE.

**Figure 4 foods-11-02391-f004:**
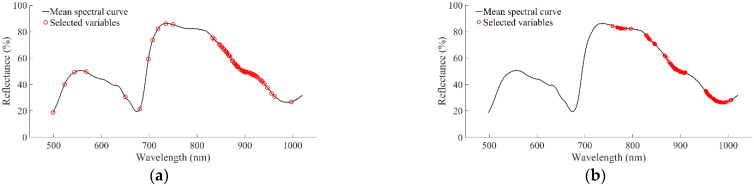
Distribution of characteristic wavelengths. (**a**) SPA; (**b**) MCUVE.

**Figure 5 foods-11-02391-f005:**
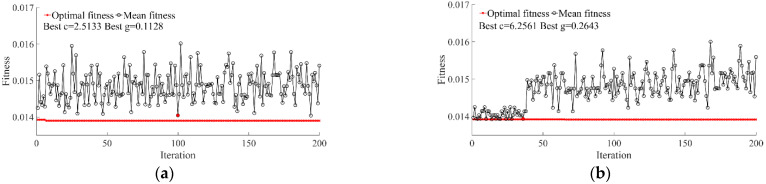
Optimization of SVR parameters. (**a**) SPA; (**b**) MCUVE.

**Figure 6 foods-11-02391-f006:**
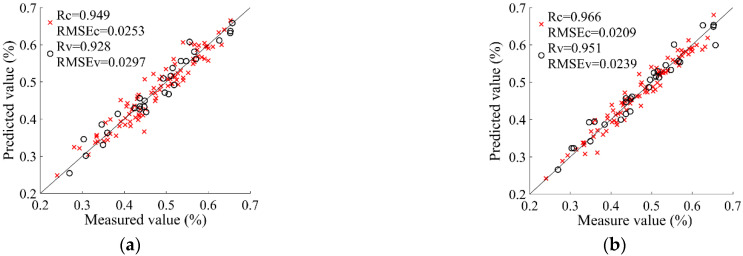
Scatter plot of the calibration set (×) and verification set (o) of stone cell content. (**a**) SPA; (**b**) MCUVE.

**Table 1 foods-11-02391-t001:** Statistics of SCC in Cs and Vs.

Sample Set	Numbers	Min (%)	Max (%)	Mean (%)	SD (%)	*p*
Cs	90	0.240	0.657	0.486	0.100	0.008
Vs	30	0.315	0.652	0.481	0.083	

Max: the maximum value of the dataset; Min: the minimum value of the dataset; Cs: calibration sets; Vs: validation sets.

**Table 2 foods-11-02391-t002:** Correlation coefficients of Cs and Vs with different S-G parameters.

	Frame Size	None	3	5	7	9
Fitting Order						
none		0.8613				
		0.8214				
1			0.8276	0.7867	0.7403	0.7012
			0.8007	0.7616	0.7150	0.6710
2				0.8306	0.7928	0.7789
				0.8035	0.7710	0.7458
3				0.8414	0.8227	0.8023
				0.8137	0.8006	0.7853
4					0.8527	0.8419
					0.8195	0.8059
5					0.8926	0.8647
					0.8210	0.8100
6						0.8589
						0.8128
7						0.8527
						0.8026

Correlation coefficients on the top and the bottom of different combinations of frame size and fitting order refer to correlation coefficients of validation set and calibration set, respectively.

**Table 3 foods-11-02391-t003:** Evaluation of PLSR based on different spectral preprocessing algorithms.

Parameter	Preprocessing Algorithm	Factor Number	R_C_	RMSE_C_ (%)	R_V_	RMSE_V_ (%)
Stone cell content (%)	None	9	0.8613	0.0360	0.8214	0.0412
	MSC	10	0.9191	0.0277	0.8879	0.0325
	SNV	10	0.9189	0.0277	0.8935	0.0315
	S-G_(7, 5)_	10	0.8926	0.0319	0.8210	0.0409
	S-G_(7, 5)_& MSC	10	0.9001	0.0308	0.8614	0.0361
	S-G_(7, 5)_& SNV	10	0.8999	0.0308	0.8641	0.0356

R_C_: the correlation coefficient of the calibration set; RMSE_C_: root mean square error of the calibration set; R_V_: the correlation coefficient of the validation set; RMSE_V_: root mean square error of the validation set; S-G(7,5): Savitzky–Golay filter with a frame size of 7 and fitting order of 5.

## Data Availability

The data presented in this study are available on request from the corresponding author. The data are not publicly available due to the request of funding scientific research projects.

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
