# Peer review of "Intelligent Evaluation of Stone Cell Content of Korla Fragrant Pears by Vis/NIR Reflection Spectroscopy"

_foods, 2022, doi:10.3390/foods11162391_

Round 1

Reviewer 1 Report

The manuscript entitled “Intelligent evaluation of stone cell content of Korla fragrant pears by Vis/NIR spectroscopy" by Wang, et al. described a Vis/NIR spectroscopy method based on several algorithms to evaluate stone cell content (SCC) of Korla fragrant pears. The manuscript appears to be mostly well-written but please, careful consider following comments:

Major concerns

1.     There is no discussion in the introduction of the significance of SCC and hence the motivation for this work other than a mere scientific journey of developing algorithms.

2.     In the Results ad Discussion sections, one would expect the authors would present a compelling correlation between their results and SCC: this is lacking. This makes it difficult to see this work beyond a mere computational/scientific adventure with no real practical significance.

3.     The authors present some characteristic functional group absorption bands, however, there is no correlation between the identified functional groups and SCC. This again is a pointer to the fact that this work cannot be seen beyond a mere computational exercise.

4.     The authors mentioned on line 212 of this paper that “Spectra in the range of 498-1020 nm were considered effective”. What exactly does this mean and how did you come to this conclusion?

5.     The sentence in line 226 of the manuscript is a conjecture, which suggests the authors are not sure of the specific acquisition settings used in the work.

6.     The authors claim their method is non-destructive: I disagree: their sample preparation renders the pearl samples unrecoverable. The best they can say is that the acquisition method and therefore, the determination of SCC is in situ.

Minor comments

7.     Lines 12 and 296: angel should be changed to angle.

8.     Please make other corrections as highlighted below.

Lines 12 and 296: angel should be changed to angle

Line 42: not been searched should be changed to not been researched

Lines 92,96,98: There should be no spacing between number and “degree” symbol

Line 221: “non-using”? This a wrong choice of words. This needs to be fixed.

Reviewer 2 Report

The article corresponding to the ref. foods-1823060 (entitled "Intelligent evaluation of stone cell content of Korla fragrant pears by Vis/NIR spectroscopy”) aims to design a Vis/NIR spectroscopic approach to measure stone cell content of pears. The work is based on the development of appropriate analytical methodology and is of quite interest for the academy and the pear cultivation. The study is well defined, and the results, in the most of case, are presented clearly.

Introduction:

Line 66: Please provide the full names of C and g.

Results:

A significant test is necessarily required in Table 1 and Table 2, especially when the data of different methods are compared.

Line 212: Please explain the reason why “Spectra in the range of 498-1020 nm were considered effective”.

Line 213-214: “Two absorption peaks were near 680 nm and 980 nm and two reflection valleys were near 550 nm and 750 nm” in Figure 2. Is it correct? It seems that 680 nm and 980 nm are the valleys, 550 nm and 750 nm are the peaks.

Line 214-216: “The peak near 550 nm and valley near 680 nm were mainly related to chlorophyll and anthocyanin in the sample epidermis”. The sentence is confusing, since 550 nm is commonly applied for analyzing anthocyanins, and 680 nm is used for chlorophyll. Please rephrase it.    

Line 265-267: “The optimal values of g were 0.2643 and 0.1128 for the first and second group, respectively”. According to Figure 5, should 0.2643 be the g value for the second group and 0.1128 for the first group? Please correct it.

Discussion:

The section needs to be re-written. It requires a comparison between Vis/NIR spectroscopy methods from the previous research and the one developed in the present study. According to the Introduction section, there are plenty of previous papers related to this topic. However, the authors provide only a summary of their results, which seems to be the conclusion rather than discussion.  

Reviewer 3 Report

The following paper : “Intelligent evaluation of stone cell content of Korla fragrant 2 pears by Vis/NIR spectroscopy” submitted to Foods deals with the use of Vis-NIRS to evaluate stone cell content of pear.

The reviewer suggests the following changes:

1- Line 12: change “angel” to “angle"

2- Line 53: it is unreasonable or reasonable?

3- Line 66: please define parameters C and g.

4- Line 73: Please give the chemical composition of the cleaning agent.

5- Line 101: “was placed on the sample until …” Please compete the sentence.

6- Line 237: please correct RMSEC to REMSEV

7- A validation section should be added where detection limit, specificity and the robustness of the method should be determined.

8- For the limit of detection: what is the minimum SCC detected by this method?

9- For the specificity: batch effect should be addressed. If a pear belonging to another batch is analyzed, does it fall within the model?

10- Effect of environment variables on the signal should be studied. What is the effect of different position of the sample? What is the effect of the distance between the probe and the sample? What is the effect of scanning and integration times?

11- Figure 2: please add the chemical band assignment information on the figure.

Reviewer 4 Report

The authors report the results of analyzes on the relationship between the spectrum of light diffused on the surface of the fruit (pear) and the content of stone particles in the flesh. Two types of work are described: laboratory determinations of the percentage of particles in the flesh by weight and numerical analyzes of the spectra of light traveling from the surface of a pear illuminated with white light. The applied numerical procedures for the analysis of light spectra seem to be effective for the evaluation of the quality of the tested pear fruit. The authors inform that in view of obtaining satisfactory research results, they plan to continue them in order to develop an operational method of fruit quality assessment, i.e. to create a device that will quickly quantify the quality of the fruit.

Two fixes needed: Lines 12 and 296  angel -> angle

Please consider whether such an article title would be better: "Intelligent evaluation of stone cell content of Korla fragrant pears by Vis/NIR reflection spectroscopy".

Round 2

Reviewer 1 Report

The authors appear to have revised the manuscript mostly based on my comments. Although there are few uncorrected errors: e.g. 

Line 221: “non-using”? This a wrong choice of words. This needs to be fixed.

Reviewer 2 Report

The authors have answered most of questions and have corrected the errors in the paper. 

Reviewer 3 Report

The authors took into account the comments of the reviewer.

I recommend the publication of the paper.

Regards.